# MITO39: Efficacy and Tolerability of Pegylated Liposomal Doxorubicin (PLD)–Trabectedin in the Treatment of Relapsed Ovarian Cancer after Maintenance Therapy with PARP Inhibitors—A Multicenter Italian Trial in Ovarian Cancer Observational Case-Control Study

**DOI:** 10.3390/cancers16010041

**Published:** 2023-12-20

**Authors:** Margherita Turinetto, Andrea Ricotti, Claudia Marchetti, Carmela Pisano, Claudio Zamagni, Chiara Cassani, Paola Malaguti, Alessandra Baldoni, Paolo Scollo, Giuseppa Scandurra, Alessandro Parisi, Grazia Artioli, Innocenza Palaia, Laura Vertechy, Alice Bergamini, Elisa Picardo, Valentina Tuninetti, Giulia Scotto, Giovanni Scambia, Sandro Pignata, Giorgio Valabrega

**Affiliations:** 1Department of Oncology, University of Turin, 10124 Turin, Italy; giulia.scotto@unito.it (G.S.); giorgio.valabrega@unito.it (G.V.); 2Clinical Trial, Ordine Mauriziano Hospital, 10128 Turin, Italy; aricotti@mauriziano.it; 3Department of Woman and Child Health and Public Health, Fondazione Policlinico Universitario A. Gemelli IRCCS, 00168 Rome, Italy; clamarchetti81@gmail.com (C.M.); laura.vertechy@policlinicogemelli.it (L.V.); giovanni.scambia@policlinicogemelli.it (G.S.); 4Department of Urology and Gynecology, National Cancer Institute IRCCS Fondazione G. Pascale, 80131 Naples, Italy; c.pisano@istitutotumori.na.it (C.P.); s.pignata@istitutotumori.na.it (S.P.); 5Addarii Medical Oncology, Istituto di Ricerca e Cura a Carattere Scientifico (IRCCS), Azienda Ospedaliero-Universitaria di Bologna, 40138 Bologna, Italy; claudio.zamagni@aosp.bo.it; 6Unit of Obstetrics and Gynecology, IRCCS Matteo Foundation, Department of Clinical, Surgical, Diagnostic and Pediatric Sciences, University of Pavia, 27100 Pavia, Italy; ch.cassani@smatteo.pv.it; 7Medical Oncology 1, Regina Elena National Cancer Institute–IRCCS, 00144 Rome, Italy; paola.malaguti@ifo.gov.it; 8Oncology and Hematology Department, Mirano AULSS3 Serenissima, 30035 Mirano, Italy; alessandrabaldoni@hotmail.com; 9Division of Gynecology and Obstetrics, Maternal and Child Department, Cannizzaro Hospital, Kore University Enna, 94100 Enna, Italy; paolo.scolllo@unikore.it; 10Medical Oncology Unit, Cannizzaro Hospital, 95126 Catania, Italy; giusy.scandurra@gmail.com; 11Clinica Oncologica e Centro Regionale di Genetica Oncologica, Azienda Ospedaliero-Universitaria delle Marche, Università Politecnica delle Marche, 60126 Ancona, Italy; alexparisi@hotmail.it; 12Department of Life, Health and Environmental Sciences, University of L’Aquila, 67100 L’Aquila, Italy; 13Oncology Department, Ca’ Foncello Hospital, 31100 Treviso, Italy; grazia.artioli@yahoo.it; 14Department of Obstetrics and Gynecology, Sapienza University, 00185 Rome, Italy; innocenza.palaia@uniroma1.it; 15Department of Gynaecology and Obstetrics, IRCCS San Raffaele Scientific Institute, Vita-Salute San Raffaele University, 20132 Milan, Italy; bergamini.alice@unisr.it; 16Obstetrics and Gynaecology 4, Sant’Anna Hospital, AOU Città della Salute e della Scienza of Turin, 88100 Turin, Italy; elisa.picardo@gmail.com; 17Medical Oncology, Ordine Mauriziano Hospital, 10128 Turin, Italy; dr.ssatuninettivalentina@gmail.com; 18Department of Life Sciences and Public Health, Università Cattolica del Sacro Cuore, 00168 Rome, Italy

**Keywords:** ovarian cancer, PLD-Trabectedin, PARPi, platinum-sensitive disease

## Abstract

**Simple Summary:**

This multicenter, retrospective analysis had the objective of comparing the efficacy of PLD-Trabectedin in patients who had already been treated with PARP-I (cases) before vs. PARPi-naïve patients (controls). Data from 166 patients were collected, composed of 109 cases and 57 controls. In total, 135 patients were included in our analyses, composing 46 controls and 89 cases. We found a median PFS of 11 months (95% IC 10–12) in the control group vs. 8 months (95% IC 6–9) in the case group (*p* value 0.0017). The clinical benefit rate was evaluated, with an HR for progression of 2.55 (1.28–5.06) for the case group (*p* value 0.008) persisting when adjusted for BRCA mutation. The study showed a statistically significant difference in terms of PFS, suggesting that a previous exposure to PARP-i might inhibit the efficacy of PLD-Trabectedin. Regarding tolerability, no remarkable disparity was noted.

**Abstract:**

Objective: While PLD-Trabectedin is an approved treatment for relapsed platinum-sensitive ovarian cancer, its efficacy and tolerability has so far not been tested extensively in patients who progress after poly ADP-ribose polymerase inhibitor (PARPi) treatment. Methodology: This multicenter, retrospective analysis had the objective of comparing patients receiving PLD-Trabectedin after being treated with PARP-I (cases) with PARPi-naïve patients. Descriptive and survival analyses were performed for each group. Results: Data from 166 patients were collected, composed of 109 cases and 57 controls. In total, 135 patients were included in our analyses, composing 46 controls and 89 cases. The median PFS was 11 months (95% IC 10–12) in the control group vs. 8 months (95% IC 6–9) in the case group (*p* value 0.0017). The clinical benefit rate was evaluated, with an HR for progression of 2.55 (1.28–5.06) for the case group (*p* value 0.008), persisting when adjusted for BRCA and line with treatment. We compared hematological toxicity, gastro-intestinal toxicity, hand–foot syndrome (HFS), fatigue, and liver toxicity, and no statistically significant disparity was noted, except for HFS with a *p* value of 0.006. The distribution of G3 and G4 toxicities was also equally represented. Conclusion: The MITO39 study showed a statistically significant difference in terms of PFS, suggesting that previous exposure to PARPi might inhibit the efficacy of PLD-Trabectedin. Regarding tolerability, no remarkable disparity was noted; PLD-Trabectedin was confirmed to be a well-tolerated scheme in both groups. To our knowledge, these are the first data regarding this topic, which we deem to be of great relevance in the current landscape.

## 1. Introduction

Ovarian cancer (OC) is a formidable adversary in the realm of women’s health, ranking as the seventh most common neoplasia among women. What makes it even more concerning is that it is the fourth most lethal cancer due to its tendency to elude early detection. In its nascent stages, ovarian cancer often remains asymptomatic, leading to a bleak 5-year overall survival rate of just 30% [1].

This characteristic alone underscores the dire need for more effective management strategies, particularly those that enhance early detection and treatment.

The prevailing standard of care for newly diagnosed high-grade serous ovarian cancer (HGSOC) typically involves primary or interval cytoreduction, followed by platinum-based chemotherapy. In recent years, the landscape of ovarian cancer treatment has undergone a significant transformation, catalyzed by the introduction of PARP (Poly ADP-Ribose Polymerase) inhibitors. Initially, these inhibitors were reserved for relapsed cases, but following the groundbreaking results of trials such as SOLO1 [2], PRIMA [3], and PAOLA1 [4], they have been integrated into first-line therapy regimens. PARP inhibitors have ushered in a new era of ovarian cancer treatment.

However, this advancement has brought about a fresh set of challenges. While PARP inhibitors have extended progression-free survival (PFS), the reality is that a significant proportion of patients will eventually experience disease relapse. This has given rise to a growing population of individuals for whom treatment strategies remain uncertain. Decisions regarding the subsequent lines of therapy, including whether to continue with PARP inhibitors or explore alternative approaches, pose complex dilemmas for both patients and oncologists.

One intriguing avenue in the quest to address these challenges involves investigating the efficacy and tolerability of non-platinum-based chemotherapy regimens in the population of patients who have previously been exposed to PARP inhibitors. Currently, there is no definitive guidance on whether patients treated with PARP inhibitors should follow different treatment paths than those who have not received such therapy. The debate around the optimal sequence of treatments is a topic of active research and discussion.

More specifically, the efficacy and tolerability of non-platinum-based chemotherapy regimens have not been described in this population. 

The combination of pegylated liposomal Doxorubicin (PLD) and Trabectedin is currently approved for individuals with relapsed OC who have experienced a platinum-free interval of at least 6 months. The approval is rooted in the compelling results of phase III trials such as OVA3012 [5] and the Monk et al. [6] study. These findings were subsequently reinforced by INOVATYON [7] and the real-world phase IV NIMES ROC [8], which emphasized that the benefits of this therapy persisted regardless of prior exposure to bevacizumab, another commonly used drug in the management of ovarian cancer.

To further elucidate the potential benefits of PLD-Trabectedin therapy, the TRAMANT-01 study, identified by its EUDRACT number 2017-000987-14, is currently underway. This trial seeks to determine whether a maintenance therapy involving PLD-Trabectedin, in comparison to Trabectedin alone, could be advantageous for patients who achieve at least a partial response after completing six cycles of combination therapy.

The trial results will be important, as a maintenance therapy with Trabectedin on its own, if proven statistically significant, will aide in providing a possible maintenance therapy in patients who have already been treated with Bevacizumab and PARPi and might not have an alternative maintenance option. These investigations into alternative therapies are a testament to the dynamic and ever-evolving nature of ovarian cancer treatment.

In response to the urgency of generating more data and insights in this domain, the Multicenter Italian Trials in Ovarian Cancer (MITO) initiated the MITO39 study. This retrospective analysis was designed to bridge the knowledge gap regarding the use of pegylated liposomal Doxorubicin (PLD) and Trabectedin, both platinum-free treatment options, in patients previously exposed to PARP inhibitors as opposed to those who had not been exposed, which were included in registrative trials. By comparing the efficacy and tolerability of these treatments in these two patient populations, the MITO39 study aims to generate a hypothesis in this still unclear scenario that can help tailor treatment strategies for individual patients and ultimately improve their chances of survival and quality of life.

## 2. Materials and Methods

The MITO39 study was conducted across multiple MITO centers, ensuring that patients were treated by experienced gynecological oncologists adhering to the current standards of care. The cohorts were drawn from cases treated at participating centers between 2009 and 2022, following the established inclusion and exclusion criteria.

This study was conducted according to the guidelines of the Declaration of Helsinki and approved by the Clinical Research Council of IRCCs Candiolo, Italy (date of approval 21 February 2021), as well as the Ethical Committee of Ospedale Mauriziano (date of approval 15 September 2022, code of approval 382/2022). Data were retrospectively collected from 166 cases, composed of 109 cases—patients subjected to PLD-Trabectedin after PARP inhibitors—and 57 controls—patients subjected to PLD-Trabectedin without ever being treated with PARP inhibitors.

The number of patients accrued was based on the maximal effort made by the centers that have participated, who have gathered all patients treated at their facility between 2009 and 2022 corresponding to our previously established inclusion and exclusion criteria. The criteria called for patients with a confirmed diagnosis of advanced epithelial ovarian cancer and known BRCA status who had been treated, according to standard practice, with PLD-Trabectedin, with or without previous exposure to PARP inhibitors.

The primary focus of this study was to compare the efficacy and tolerability of PLD-Trabectedin between the two patient groups. Additionally, we also collected data on various types of toxicity, including hematological, gastro-intestinal, hand–foot syndrome, fatigue, and liver toxicity. Follow-up data were tracked until November 2022 to provide a comprehensive overview of patient outcomes.

Demographic and clinical data were retrospectively retrieved from dedicated databases at each institution. The database elements included: patient characteristics at initial diagnosis (demographics, tumor stage according to the International Federation of Gynecology and Obstetrics [FIGO] criteria, histology, year of diagnosis), BRCA status, surgical, chemotherapy and PARPi regimen details, number and type of chemotherapy regimens performed after PARPi, PLD-Trabectedin toxicity details, and PFS for PLD-Trabectedin.

Statistically significant differences were present for the BRCA status, as well as for the stage at diagnosis; while we deemed significant for our study the difference in BRCA status and therefore carried out a multivariate analysis, we did not think that the difference in initial staging would influence the response to chemotherapy in a multi-treated metastatic population.

Patients were divided into the two study groups, and descriptive and survival analyses were performed for each. Due to missing information, 135 patients were included in our analyses, composed of 46 controls and 89 cases. The populations were compared for categorial variables (age, stage at diagnosis, type of surgery at diagnosis, histology), and no statistical difference was found (Table 1).

Clinical response to therapy was evaluated using the modified Response Evaluation Criteria in Solid Tumor (RECIST) version 1.1.

Progression-free survival (PFS) was defined as the time between the date that therapy started and the date of disease progression or death or last contact. At the time of the data cut off, all but one patient had experienced progression of disease (PD).

The results are presented as median and interquartile range [IQR] for continuous variables and number and percentage for categorical ones. A comparison of variables between the patients treated with PLD-Trabectedin and the control group was performed using the Mann–Whitney or Fisher’s exact test, when appropriate.

PFS was estimated using the Kaplan–Meier method and compared with the log-rank test. Based on clinical assumptions, a multivariate Cox model was performed to describe the treatment risk factor associated with PFS adjusted for BRCA and age; the model was stratified by line of treatment in order to account for different basal risks. The proportional hazard assumption was checked using the Schoenfeld residuals. A *p*-value less than 0.05 was considered statistically significant. All statistical analysis were performed using R version 4.2.1. Based on clinical assumptions, a multivariate analysis using the Cox model was carried out on PARPi and BRCA data to identify the risk of PFS.

We also decided to stratify our population based on the line at which PLD-Trabectedin was administered, dividing between 1st, 2nd, 3rd, and >4th line, as we deemed the timing of PLD-Trabectedin as a potential confounding element.

Details of the PARPi regimens administered are outlined in Table 2.

## 3. Results

Median PFS was 11 months (95% IC 10–12) in the control group vs. 8 months (95% IC 6–9) in the case group (*p* value 0.0017) (Figure 1).

The clinical benefit rate (CBR) was evaluated as well, with an HR for progression of 2.55 (1.28–5.06), *p* value 0.008) for the case group, calculated based on the stratification of the line of treatment. It also persisted when adjusted for the statistically different presence of BRCA mutations in the two groups.

Regarding safety, we compared hematological toxicity, gastro-intestinal (GI) toxicity, hand–foot syndrome (HFS), fatigue, and liver toxicity; no statistically significant disparity was noted, except for HFS: seven patients suffered from HFS (3 CTCAE G1, 3 CTCAE G2, 1 CTCAE G4) in the controls compared to one G2 in the cases (*p* value 0.006) (Table 3).

We also looked at the distribution of G3 and G4 toxicities, which were equally represented.

## 4. Discussion

PARP inhibitors in platinum-sensitive disease in clinical practice have revolutionized the treatment of OC, with most patients undergoing at least a partial response to platinum therapy, receiving it as a maintenance therapy in clinical practice.

Although astounding, even in the best-case scenario of a BRCA-mutated patient receiving Olaparib as a first-line therapy, the updated analyses at 5 years of the SOLO1 trial reported a PFS of 48% (95% CI 41–55), which would mean that 52% had relapsed and had needed further treatment [2]. 

Thus, exposure to PARP inhibitors has selected a population with fundamental differences compared to those represented in all the registrative trials of the chemotherapy lines currently approved for OC; however, this has now changed and all current trials account for PARP-inhibitors-treated patients; this brings about the question of whether the chemotherapy regimens we have used previously have the same efficacy and tolerability on tumors whose microenvironment has been changed by the interaction with PARP inhibitors, which has shown to have a great impact [9,10,11]. 

Currently, there are a variety of possible approaches to a patient who progresses on or after PARP inhibitors [12,12]. Although a solid therapeutic algorithm is still missing, the use of loco-regional treatments such as surgical intervention or radiotherapy (RT) and the continuation of PARP inhibitors therapy is widely spread in oligo-metastatic progression, with data so far coming only from retrospective evidence alone [13,14]; therefore, the validation of this approach through a randomized prospective trial is needed.

In case of a progression not amenable to surgical or radiation treatment, a further line of chemotherapy is recommended based on the platinum-free interval (PFI); in the post-PARP inhibitors era, several concerns have been raised regarding the efficacy of platinum-based therapies in these patients [15] as newly emerging data point towards a platinum resistance due to a cross-resistance mechanism [16,17]. While many efforts are ongoing on a translational level to better understand this [18,19,20,21,22,23], we are starting to gather clinical evidence [24,25]; in a post hoc analyses of the SOLO2 trial, the time to second progression (TTSP) was significantly longer in the placebo cohort than the Olaparib cohort, with a TTSP of 14.3 vs. 7.0 months (HR 2.89, 95% CI 1.73–4.82) for platinum salts, while for non-platinum-based therapies, it was 8.3 vs. 6.0 months (HR 1.58, 95% CI 0.86–2.90) [26]. 

Similarly, an Italian retrospective real life study of Olaparib [27] highlighted an ORR of only 22.2% to platinum in patients with a PFI of more than 12 months who had progressed after having received Olaparib. 

The MITO39 is the second and largest study to focus on the efficacy of non-platinum-based CT in platinum-eligible patients who progress after PARP inhibitor therapy. 

With this retrospective study, we were able to show a statistically significant difference in terms of PFS between patients previously treated with PARP inhibitors and PARP-inhibitors-naïve patients treated with PLD-Trabectedin, thus suggesting that a previous exposure to PARP inhibitors might hinder the efficacy of the regimen. Interestingly, however, the performance of PLD-Trabectedin in PARP inhibitors-exposed patients was similar to the one observed in other studies where patients were PARP-naïve.

Our numbers are in line with the previously mentioned trials; in OVA301 [5], the PFS for partially platinum-sensitive patients was 7.4 months (HR 0.65, IC 95% 0.45–0.92), in the INOVATYON [7] trial it was 7.5 months (95% IC 3.0–11.5), and in the real life European retrospective NIMES-ROC [8] it was 9.46 months (95% IC 7.9–10.9). 

Regarding tolerability, the comparison did not yield a remarkable disparity; PLD-Trabectedin was confirmed to be a well-tolerated and manageable scheme in both groups.

The present study contains both limitations and strengths. Limitations include those associated with the retrospective nature and the intrinsic risk of confounding due to the absence of randomization, the relatively small number of patients included, and the significant difference in BRCA-mutated patients between the two groups. The strengths of this study are the homogeneity of the cases enrolled, in terms of clinical features and treatment administered, and the soundproof statistical analyses conducted. 

To our knowledge, these are the first data regarding the activity of a non-platinum agent in patients treated with PARP inhibitors compared to PARP-inhibitors-naïve controls, which we deem to be of great relevance in the current era; PARP inhibitors have now been a standard of care for years, prompting the need to further explore whether exposure to this targeted therapy has any impact on later lines. This investigation aims to determine if such exposure could lead to a change in perspective when selecting the appropriate treatment for a patient who experiences a relapse after maintenance therapy.

While PLD-Trabectedin still stands as an option for platinum-sensitive relapsed disease, the evidence gathered by this study suggests that the activity of a non-platinum-based regimen might be lower than expected in patients pre-treated with PARP inhibitors, which will need to be factored in when deciding the next line of treatment for such a patient. Many different characteristics need to be considered when deciding a new strategy, and this includes previous PARPi exposure.

The observational nature of our study limits the significance of our findings, which will need to be validated further; it would also be beneficial to better understand on a molecular level whether there are drug-specific resistance mechanisms that might justify our data.

Interestingly, previous data have shown that PLD + Trabectedin after PARP inhibitors achieves similar oncological outcomes compared with a platinum-based regimen [28]. Therefore, in a sequence strategy, PLD + Trabectedin still remains a reasonable choice following PARP inhibitors progression, sparing platinum compound for subsequent recurrence. 

## 5. Conclusions

Our study highlights a significant difference in PFS between PARP-inhibitors-naïve and PARP-inhibitors-treated patients when treated with PLD-Trabectedin; while this is an observational retrospective study, we believe that these results confirm the need to address patients who relapse after PARP inhibitors differently.

## Figures and Tables

**Figure 1 cancers-16-00041-f001:**
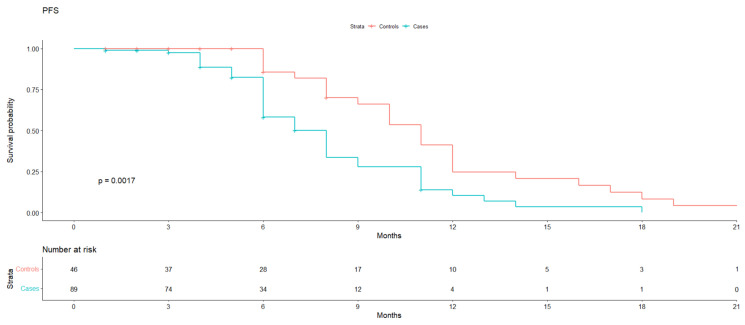
Median PFS curves.

**Table 1 cancers-16-00041-t001:** Categorial variables considered.

Variable	Controls (%)	Cases (%)	*p* Value
**N**	46	89	
**Age (Median IQR)**	60.50 [55.25, 67.00]	56.00 [49.00, 64.00]	0.006
**Surgery**			0.034
PDS	22 (47.9)	59 (66.3)	
IDS	19 (41.3)	27 (30.3)	
Not operated	3 (6.5)	3 (3.4)	
**Histology**			0.153
HGSOC	38 (82.6)	81 (91)	
Endometrioid Carcinoma	2 (4.4)	3 (3.3)	
Endometrioid + CC Carcinoma	1 (2.2)	0 (0.0)	
HGSOC + Endometrioid	0 (0.0)	1 (2.2)	
Genital Carcinoma	1 (2.2)	0 (0.0)	
Undifferentiated	2 (4.4)	1 (1.1)	
Serous-Mucinous Carcinoma	(0.0)	1 (1.1)	
Peritoneal Primary Cancer	(0.0)	1 (1.1)	
Serous + Focal Sarcomatoid Component	1 (2.2)	0 (0.0)	
Mucinous Carcinoma	1 (2.2)	0 (0.0)	
**BRCA**			0.001
WT	43 (93.5)	59 (66.3)	
BRCA1 MUT	2 (4.3)	22 (24.7)	
BRCA 2 MUT	1 (2.2)	8 (9.0)	
**Stage at Diagnosis**			0.044
I	0 (0.0)	2 (3.4)	
II	0 (0.0)	8 (9.0)	
III	35 (76.1)	66 (74.2)	
IV	11 (23.9)	12 (13.5)	

**Table 2 cancers-16-00041-t002:** Details of PARPi therapy.

N	89 (%)
**PARPI**	
OLAPARIB	31 (34.8)
RUCAPARIB	3 (2.2)
NIRAPARIB	56 (62.8)
**LINE**	
I	15 (16.9)
II	60 (68.5)
III	9 (10.1)
>IV	4 (4.5)

**Table 3 cancers-16-00041-t003:** Distribution of toxicities of PLD-Trabectedin.

Toxicities-Grades	Controls (%)	Cases (%)	*p* Value
**Anemia**			0.8
0	36 (78.3)	66 (74.2)	
I–II	7 (15.2)	16 (18)	
III–IV	3 (6.5)	7 (7.8)	
**Neutropenia**			0.78
0	29 (63.0)	48 (53.9)	
I–II	4 (8.7)	10 (11.2)	
III–IV	13 (28.2)	31 (23.9)	
**Thrombopenia**			0.9
0	42 (91.3)	76 (85.4)	
I–II	3 (6.5)	8 (9)	
III–IV	1 (2.2)	5 (5.6)	
**Gastro-INTESTINAL**			0.4
0	38 (82.6)	53 (59)	
I–II	5 (10.9)	27 (34.8)	
III–IV	3 (6.5)	9 (10.1)	
**Liver Toxicity**			0.97
0	36 (78.3)	70 (78.7)	
I–II	8 (17.3)	14 (15.7)	
III–IV	2 (4.3)	5 (5.6)	
**Fatigue**			0.69
0	40 (87.0)	68 (76.4)	
I–II	5 (10.8)	15 (16.9)	
III–IV	1 (2.2)	6 (6.7)	
**HFS**			0.006
0	39 (84.8)	88 (98.9)	
I–II	6 (13)	1 (1.1)	
III–IV	1 (2.2)	0 (0)	

## Data Availability

The data presented in this study are available on request from the corresponding author.

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
