# Peer review of "MITO39: Efficacy and Tolerability of Pegylated Liposomal Doxorubicin (PLD)–Trabectedin in the Treatment of Relapsed Ovarian Cancer after Maintenance Therapy with PARP Inhibitors—A Multicenter Italian Trial in Ovarian Cancer Observational Case-Control Study"

_cancers, 2023, doi:10.3390/cancers16010041_

Round 1

Reviewer 1 Report

Comments and Suggestions for Authors

Turinetto et al. describe in their retrospective analysis the possible efficacy of PDL-Trabectedin in the treatment of Ovarian cancer in combination with or without PARPi. I find this article interesting, filling a possible gap occurring in this field.

However, before considering the manuscript for publication there are some issues that need to be clarified:

1.       The simple summary is difficult to follow. I assume that some word in the third line are missing.

2.       The differences of the two examined groups must be better presented. From Table 1 it is obvious that we have some differences regarding tumor stage between the two groups and BRCA. The second difference is only presented in the Discussion section but I should also be commented upon also during describing the patients populations.

3.       Why did the authors not adjusted to stage in the multivariate Cox analysis. Especially when the two examined groups appear different in this context. It could be assume that the better outcome observed in the case group is attributed to having more patients with stage I or II at diagnosis.

4.       Please remove all bullets from Table 1 and 2 and arrange the words links.

5.       Regarding the multivariate survival analysis the authors should present in detail the model adjusted and not only the significant results.

6.       A Table regarding the toxicity or safety of the therapeutic intervention in both groups with the comparisons should also be added so that we could have a better view of these parameters.

Comments on the Quality of English Language

I do not find significant issues regarding the English language with an exception of the simple summary.

Author Response

Dear reviewers,

we’ve carried out the modifications required as per your suggestions, with the specifics explained below. Thank you for your valuable feedback.

  1. We’ve rephrased the simple summary
  2. At line 154 we’ve outlined the differences in the populations
  3. While we did adjust for BRCA and carried out a multivariate analyses we did not think the different stage at diagnosis would influenciate a multitreated metastatic population, furthermore the population with more initial stages at diagnosis is actually the one with the lowest PFS
  4. We’ve modified the tables
  5. We’ve presented all the analyses we’ve carried out and not only the significative ones, the prespecified objectives were PFS, CBR and toxicities
  6. It’s been added

Reviewer 2 Report

Comments and Suggestions for Authors

Clearly state the research gap that the MITO39 study aims to address, emphasizing the need for data on the efficacy and tolerability of non-platinum-based chemotherapy regimens in patients previously exposed to PARP inhibitors.

Elaborate on the relevance and significance of the ongoing TRAMANT-01 study, which investigates the potential benefits of PLD-Trabectedin maintenance therapy compared to Trabectedin alone in patients who have achieved a partial response after completing the combination therapy.

Acknowledge the limitations of the observational retrospective study design, such as potential selection biases and the absence of randomization. This will help provide a more balanced interpretation of the findings and enhance the credibility of the conclusions.

Instead of stating that the study "confirms the need to address patients who relapse after PARP inhibitor differently," emphasize that the findings suggest the potential for differential treatment approaches in patients who relapse after PARP inhibitor therapy.

Expand on the implications of the study findings for clinical decision-making. Highlight the importance of considering the treatment history of patients, including prior PARP inhibitor exposure, when determining subsequent treatment strategies. Additionally, emphasize the need for prospective studies to validate these results and explore the underlying mechanisms driving the observed differences in PFS. This will provide a more comprehensive and actionable conclusion for clinicians and researchers.

Author Response

Dear reviewers,

we’ve carried out the modifications required as per your suggestions, with the specifics explained below.

Thank you for your valuable feedback, in particular:

We’ve made sure to elaborate further on the TRAMANT study (line 117), better outlined the limitations of the study (line 269). We’ve also added your suggestions on further prospective and translation analyses (line 275)

Thank you for your time and effort

Round 2

Reviewer 1 Report

Comments and Suggestions for Authors

I do not have any suggestions for the authors.